# Enhancing Driving Safety through User Experience Evaluation of the C-ITS Mobile Application: A Case Study of the DARS Traffic Plus App in a Driving Simulator Environment

**DOI:** 10.3390/s24154948

**Published:** 2024-07-30

**Authors:** Gregor Burger, Jože Guna

**Affiliations:** 1Faculty of Electrical Engineering, University of Ljubljana, Tržaška cesta 25, 1000 Ljubljana, Slovenia; joze.guna@ntf.uni-lj.si; 2Faculty of Natural Sciences and Engineering, University of Ljubljana, Aškerčeva cesta 12, 1000 Ljubljana, Slovenia

**Keywords:** mobile application, road safety, driving simulator, eye-tracking, quantitative driving parameters, user experience, Cooperative Intelligent Transport Systems (C-ITS), DARS Traffic Plus

## Abstract

The paper evaluates the DARS Traffic Plus mobile application within a realistic driving simulator environment to assess its impact on driving safety and user experience, particularly focusing on the Cooperative Intelligent Transport Systems (C-ITS). The study is positioned within the broader context of integrating mobile technology in vehicular environments to enhance road safety by informing drivers about potential hazards in real time. A combination of experimental methods was employed, including a standardised user experience questionnaire (meCUE 2.0), measuring quantitative driving parameters and eye-tracking data within a driving simulator, and post-experiment interviews. The results indicate that the mobile application significantly improved drivers’ safety perception, particularly when notifications about hazardous locations were received. Notifications displayed at the top of the mobile screen with auditory cues were deemed most effective. The study concludes that mobile applications like DARS Traffic Plus can play a crucial role in enhancing road safety by effectively communicating hazards to drivers, thereby potentially reducing road accidents and improving overall traffic safety. Screen viewing was kept below the safety threshold, affirming the app’s efficacy in delivering crucial information without distraction. These findings support the integration of C-ITS functionalities into mobile applications as a means to augment older vehicle technologies and extend the safety benefits to a broader user base.

## 1. Introduction

Mobile phones and mobile applications are widely used in mobility, often in road vehicles as entertainment or a navigational aid. They can benefit the driver; however, they can provide distractions and possibly impact driving safety as well. Interaction with mobile phones, similar to interaction with infotainment systems or navigational devices in vehicles, distracts drivers from looking at the road and following the traffic activity. In this study, we investigated the impact of the DARS Traffic Plus mobile application on driver safety and user experience within a realistic driving simulator environment.

The European Member States and road operators have developed the C-Roads Platform [1] devoted to testing and implementing Cooperative Intelligent Transport Systems (C-ITS) services [2] with the aim of cross-border harmonisation and interoperability. The C-ITS enables vehicles to interact directly with each other and the surrounding road infrastructure. The C-ITS typically involves vehicle-to-vehicle (V2V) and vehicle-to-infrastructure (V2I) communication in road transport. There are ongoing, long-lasting E.U. research project activities and significant financial funding to provide real-world solutions for V2V and V2I issues in the Cooperative Intelligent Transport Systems and its services. C-ITS services provide drivers with information about upcoming potentially hazardous traffic events on the road. Received notifications can impact drivers and distract them from driving safely. Our research presents the impact of drivers’ interactions with C-ITS services and the user experience that they provide.

The Slovenian National Motorways operator, Družba za avtoceste v Republiki Sloveniji (DARS d.d.) [3], introduced a DARS Traffic Plus mobile application targeting public drivers. The application aims to provide accurate and real-time C-ITS information to drivers and help them be more informed and, therefore, safer. 

We conducted a user experience evaluation of the DARS Traffic Plus mobile application in a driving simulator. Our research focused on examining the impact of using mobile phone applications to provide and report information about hazardous locations while driving on the road and how information about hazardous locations influences the perceived drivers’ sense of safety. We examined the usability of different visual and sound combinations of notifications for interacting with the driver to provide clear and precise information about hazardous driving events. The user experience and usability of the DARS Traffic Plus mobile application were evaluated to further enhance the experience. The driving simulator results show that drivers who are informed about hazardous driving locations readjust their driving style, which presents an important behaviour change. The average mean time of continuous looking at the mobile phone screen does not exceed 2 s, suggesting mobile phone usage does not provide a distraction while driving.

The key findings of this paper include the following:The application significantly enhanced drivers’ perceptions of safety when they received notifications about hazardous locations.Drivers preferred receiving notifications at the top of the mobile screen accompanied by a beep, suggesting that notification delivery methods critically influence user experience and safety perception.The application maintained driver attention within safe limits, with an average screen viewing time of 1.64 s, below the critical 2-s threshold suggested by established safety guidelines.

## 2. Literature Review

We conducted a literature review to address the works related to research on using mobile phones and mobile applications while driving. Additionally, we reviewed the research on interaction with mobile phones, the influence of mobile phones on driver awareness, and distracted driving in a driving simulator.

A driving simulator presents a safe, practical, and controlled environment for driving evaluations with a correlation between real-life driving and simulated driving [4]. The validity of the assessments performed in driving simulators depends on the simulators’ fidelity to physical hardware and the approaches used to evaluate driving performance in driving simulators compared to driving in the real world [5,6].

Distracted driving and performance measures for distracted driving have been of substantial interest in recent research [7]. Driving performance parameters used in driving simulator research include lateral control, longitudinal control, reaction time, gap acceptance, eye movement, and workload measures. Mobile phones are one of the most common distractions while driving, as identified in research [7]. In addition to driving performance parameters from driving simulators and eye-tracking data [8], other typical methods used in driving simulator research are validated questionnaires [4] and open-ended questions and interviews, which are typically used to measure drivers’ perceptions and awareness [9]. 

The National Highway Traffic Safety Administration (NHTSA) recommends a 2-s limit for glances away from the roadway [10]. Their research examined the impact of interacting with two infotainment systems, Android Auto and Apple CarPlay. Participants in the study met the criteria when using voice control but failed to meet the criteria for most tasks when using touch controls for both systems. Not all interactions with mobile applications in the car while driving have a negative notation, however [11]. A research study found the following desirable types of smartphone applications where safety considerations have the same level of importance as concerns: collision warning, texting prevention, voice control for text-to-speech and giving commands, and in-vehicle data recorder (IVDR) systems [11]. Nevertheless, when in-vehicle information systems (IVIS) are designed to be safe, they can still distract drivers from driving. The distraction increases with prolonged task completion time in tasks like reading or text entry. Another study suggested that the time needed to complete the task should be shortened to reduce the possible distraction of the driver [12].

User interfaces are important for the safety and acceptance of systems [13,14,15] and touchscreen interfaces [16] in cars. One case where user interfaces and notifications in cars are important for safety is automated vehicles. Automated vehicles necessitate driver intervention during emergencies, emphasising the critical role of the human–machine interface (HMI) [17]. No standard guidelines exist for automated driving interfaces, but a heuristic evaluation of typical use cases suggests that HMI elements should be functionally grouped to aid mode recognition. Visual interfaces need clear luminance and colour contrast to convey messages in a user-friendly language and concise text. A maximum of five colours (excluding white and black) is recommended for coding system states. Additionally, research on mobile-delivered variable speed limits with distance-based alerts shows improved traffic safety by enabling quicker speed adjustments and smoother decelerations, enhancing driver awareness [18].

Another case for user interfaces and notifications in cars is automatic driving systems (ADS). In ADS, a takeover request (TOR) plays a fundamental role, where drivers must take control of the vehicle on sudden notice. A literature review [19] noted varying times to complete a TOR. One source suggested that a complete TOR should be performed in 7 s, while other sources suggested that this process should be performed in 2.5 and 3.5 s, respectively.

The third case is the C-ITS. The C-ITS is not only active in Europe but also in Australia. The Ipswich Connected Vehicle Pilot in Australia [20], discussed in a methodology evaluation publication [21], suggested that there is a significant difference in the approaching speed profile in cases of driving with the human–machine interface engaged compared to cases of driving without the human–machine interface engaged.

The accommodation or adaptation period at the beginning of the evaluation in a driving simulator is crucial for the experiment outcome [22]. There are two predominant techniques for adapting to a driving simulator. The first is driving for a predefined fixed time, and the second is using subjective sensations of adaptation [23]. The authors examined the different complexity and demands of the roads, time adaptation, and participants’ subjective sensations of acclimatisation. The results showed that the more demanding curved roads required longer adaptation times. The subjective sensation of acclimatisation was in line with most performance measures for all road types. 

Eye-tracking systems are essential to the driving parameters measured in a driving simulator [24]. However, their role is not limited to the measurement of driving parameters. Eye-tracking systems are used to evaluate user experience (UX) and usability [25,26,27,28]. User experience is measured by several concepts: experience, emotion, affection, aesthetics, and persuasion [29]. User interfaces are important for the safety and acceptance of systems [13,14,15] and touchscreen interfaces [16] in cars.

## 3. Materials and Methods

### 3.1. Participants

In a driving simulator study evaluation with the DARS Traffic Plus mobile application version 3.0 (hereafter referred to as the DT+ mobile application), 39 individuals participated. Participants in the simulator study were selected among pre-existing DT+ mobile application users and volunteers who responded to the study invitation. 

Nine participants had previous experience with the DT+ mobile application; some were aware of the DT+ mobile application, and others did not know about the DT+ mobile application. None of the participants had previous experience with a professional driving simulator, and five had experience in computer games. 

All participants were healthy, had suitable eyesight for driving, and had a valid driving license for driving a personal vehicle. Only one participant was a professional freight vehicle operator. 

Participants signed written informed consent before taking part in the simulator study evaluation. After completing the simulator study evaluation, participants were awarded a small-value promotional gift from the DARS company.

### 3.2. Apparatus

The driving simulator study evaluation with the DT+ mobile application was performed in an official professional driving simulator (Figure 1) used by the DARS company to assess the existing highway road network efficiency and future road network development. The simulator resembled the latest version of the Renault Clio’s interior and had all the road-legal vehicle functionalities. Additionally, the simulator was fitted with a professional eye-tracking system, Smart-eye pro [30] version 8.0, with two cameras (Smart Eye, Gothenburg, Sweden). The sampling rate of the eye-tracking system was set to 60 Hz. We used a Samsung Galaxy S8 mobile phone (Samsung, Suwon, Republic of Korea) to run a DT+ mobile application. We recorded the quantitative driving simulator parameters, eye-tracking data, and participant driving recordings on an external computer. A speaker with a subwoofer simulated the vehicle engine and environment sounds in the driving simulator. We carefully cleaned the driving simulator, mobile phone, pen, table, and chair with wet tissues after each participant.

### 3.3. Driving Simulator Scenarios

We prepared four driving scenarios for the driving simulator study. Scenarios S0, S1, S2, and S3 presented different tasks in the driving simulator, as listed in Table 1. Scenarios included under the standard Hazardous Locations Notification category were Accident Zone Warning, Traffic Jam Ahead Warning, Obstacle on the Road Warning, and Weather Condition Warning. All participants were instructed to drive according to traffic regulations.

Figure 2 presents the diagram of Scenario S2. On the diagram, we marked the start position of the evaluation, driving directions, including the positions of the Hazardous Locations Notification warning events, and the end position of the evaluation. The diagram of the scenario was then implemented into the driving simulator programme used to evaluate the DT+ mobile application. We designed four diagrams, one for each driving scenario.

### 3.4. Metrics

We used a validated meCUE 2.0 Questionnaire [31] to evaluate the user experience. A specific questionnaire was used in the post-interview after the driving scenarios, evaluating participants’ overall experience and driving habits.

#### 3.4.1. meCUE 2.0 Questionnaire

The meCUE 2.0 Questionnaire is based on the analytical component model of user experience that grants a modular evaluation of the central aspects of user experience. The questionnaire consists of five modules:Module I Perception of Instrumental QualitiesModule II Perception of Non-Instrumental QualitiesModule III User EmotionsModule IV Consequences of UseModule V Overall Evaluation

Items are scored on a seven-point scale from 1 to 7. A value of 1 represents the “strongly disagree” response, and a value of 7 represents the “strongly agree” response. An exception is the Module V Overall Evaluation, varying from values of −5 to 5 in a 0.5 interval. Module scores are calculated as arithmetic values.

#### 3.4.2. Post-Interview

We conducted a post-interview at the end of the driving simulator study. The post-interview was designed to evaluate the DT+ mobile application user experience, the clarity of icons, the colour palette, the text size, and similar graphical design elements. Another important aspect was the participant’s feeling of safety when reporting or receiving the Hazardous Locations Notification events while driving. We used a five-point Likert scale. The value of 1 represented the “strongly disagree” response, while the value of 5 represented the “strongly agree” response.

#### 3.4.3. Experiment Environment

The driving simulator study evaluation with the DT+ mobile application was conducted in a controlled environment. The environmental conditions were constant, did not change during the study evaluation, and were without outside disturbances. The experimenters controlled the temperature, noise level, lighting conditions, and humidity. 

#### 3.4.4. Evaluation Procedure

The driving simulator study was performed using a repeated measured within-subjects design; participants experienced all four scenarios in one session. The evaluation was balanced as half of the participants completed Scenario S1 (without DT+ mobile application) first, and the other half completed Scenario S2 (with DT+ mobile application) first. We used the Latin square design [32] principle to balance the participants. The independent variables were driving with and without the DARS Traffic Plus mobile application. All participants were instructed not to consume food, coffee, alcohol, or energy drinks up to two hours before the evaluation. The driving simulator study session lasted, on average, 90 min. While performing a driving simulator study, only one participant and two researchers were present. One researcher led the driving simulator study; the other managed the equipment and observed the study. 

When participants arrived at the diving simulator location, they were greeted and escorted to the simulator. We provided them with an introductory explanation and timetable about the evaluation procedures. We also asked them about their current sickness levels, well-being, or other health-related limitations. 

We introduced the participant to the evaluation procedure. They were informed about the study, signed written informed consent, and confirmed participation upon understanding the evaluation procedure. Initially, they completed a demographic questionnaire about personal driving style, technology usage in vehicles, and mobile applications used for driving. After the initial questionnaires, the participants familiarised themselves with the DT+ mobile application. 

After that, we invited the participants to sit in the driving simulator and showed them how to set up the correct driving position and operate the simulator. Setting up the driving simulator was followed by the calibration of the eye-tracking system, which was performed before every driving session to ensure stable eye-tracking. We designed Scenario S0 (the introduction scenario) so the participants would become familiar with driving in the simulator. Scenario S0 included no Hazardous Locations Notification events; only light vehicle traffic was present on the roads. We instructed participants to drive according to the traffic regulations and traffic signs included in the scenario. During the duration of the driving scenarios, we also monitored the sickness levels of the participants. 

After completing Scenario S0, the participants left the simulator and filled out the meCUE 2.0 questionnaire. 

We used the counterbalancing principle in the evaluation, where half of the participants completed Scenario S1 and then Scenario S2. The other half completed Scenario S2 first and then Scenario S1. 

Participants returned to the driving simulator and received instructions for Scenario S1. Instructions for Scenario S1 stated that they must drive according to the traffic regulations and traffic signs included in the scenario. While driving, they encountered Hazardous Locations Notification events (Accident Zone, Traffic Jam Ahead, Obstacle on the Road, and Weather Condition Warning). They must safely avoid all the Hazardous Locations Notification events. After completing Scenario S1, the participants left the simulator. They filled out the new round of the meCUE 2.0 questionnaire.

Scenario S2 was a repeat of Scenario S1, with one key difference. Participants used the DT+ mobile application while driving in the simulator. The DT+ mobile application informed participants while driving about the Hazardous Locations Notification events with a series of on-screen messages. Notifications were orange or red, with noticeable icons and text explaining the meaning of the messages. After completing Scenario S2, the participants again left the simulator and filled out the meCUE 2.0 questionnaire.

Scenario S3 was designed to test the additional functionalities of the DT+ mobile application. We focused on sound notifications and different ways of presenting the messages to the driver. We tried different sounds to test for the best perception while driving, such as beep, no beep, and speech. Hazardous Locations Notification messages were coloured blue for informative messages, orange for messages warning about possible danger, and red for messages demanding drivers’ action on the road. The messages had two different ways of presentation, from the top or bottom of the mobile phone screen. Additionally, we asked participants to enter the new driving route and check the number of warning messages for their new trip. The last two tasks were designed to observe more focused interaction with the DT+ mobile application while driving. After completing Scenario S3, the participants again left the simulator for the final time and filled out the meCUE 2.0 questionnaire.

We concluded the driving simulator evaluation with a post-interview about the overall user experience of the DT+ mobile application. In the post-interview, participants were asked to assess the application’s user experience, usability, and their perception of safety when reporting or receiving Hazardous Locations Notification messages. They provided their opinions on the visibility of the messages, the best-perceived sound notifications, and the interface design features. 

#### 3.4.5. Research Questions

Q1: Does the DARS Traffic Plus (DT+) mobile application provide a good user experience for the driver? 

Q2: Does the DARS Traffic Plus (DT+) mobile application impair the driver’s driving performance? 

Q3: Does the DARS Traffic Plus (DT+) mobile application provide measurable and positive differences in driving parameters when driving? 

Q4: Does the DARS Traffic Plus (DT+) mobile application increase the feeling of safety while driving? 

## 4. Results

We used IBM SPSS packet version 25, MS Office Excel 2019, and Tableau software version 2020.4 for the data analysis. We checked for a normal distribution of the data; however, the data were not normally distributed. Therefore, we used the Wilcoxon signed-rank test.

### 4.1. Participant Demographic Data

We selected a group of real-life drivers with valid licenses for a pilot evaluation study. Thirty-nine participants took part, including 20 females and 19 males. Thirty-three participants (16 females and 17 males) completed all four driving scenarios, while six did not due to simulator sickness.

All participants were healthy and had adequate vision or corrective aids. Ages ranged from 22 to 62 years, with a mean age of 35.61 years (SD = 10.69 years). Their educational backgrounds varied, mainly in engineering or social sciences, with secondary or university education. Only one participant was a professional freight vehicle operator; the rest were non-professional drivers.

Most participants drove 10–20,000 km per year, primarily using a dynamic driving style, mainly on motorways and local roads. They treat daily commutes differently than non-daily commutes and are influenced by other drivers’ actions. 

### 4.2. Technology and Mobile Application Usage

Among the participants, 14 used iOS and 19 used Android phones. Smartphone use in the car varied: nineteen participants used a hands-free system daily for calls, four used a phone holder with a hands-free system, five held their phones while driving, four placed their phones on the driver’s seat, and one did not use a mobile phone while driving.

Google Maps was the most used app for driving (32 participants), followed by the DT+ app (9 participants) and in-car navigation systems (7 participants). Eight participants were existing DT+ users, while twenty-five were new. Key DT+ features included congestion alerts (8 participants), traffic notifications (8 participants), and road condition checks (7 participants).

Cruise control was the most used safety system (17 participants), followed by adaptive cruise control (5 participants) and blind spot detection (5 participants). 

### 4.3. Usability and User Experience Insights

We compared modular evaluations of questionnaire subscales for two driving scenarios in the driving simulator. The first is Scenario S1, where participants drove without the DT+ mobile application, and the second is Scenario S2, where participants drove with the DT+ mobile application. Results are presented in Table 2 and Figure 3.

The user experience evaluation of the DT+ mobile application was performed through driving. Scenario S1 is similar to Scenario S2, with the support of the DT+ mobile application in Scenario S2. Participants rated their driving experience with a user experience questionnaire. The difference between the scores of the questionnaires in Scenarios S1 and S2 is the user experience of the DT+ mobile application. In our case, the scores for the user experience questionnaire were higher for Scenario S2. Therefore, the DT+ mobile application brings a positive user experience to driving.

For Module I, the Scenario S1 Usefulness scale received a 5.30 (SD = 1.02) score, while the Usefulness scale in Scenario S2 received a 5.75 (SD = 0.63). The Usability scale in Scenario S1 received a score of 5.80 (SD = 0.85), while in Scenario S2, the Usability scale received a score of 5.95 (SD = 0.62).

The Visual aesthetic, Status, and Commitment scales in Module II were lower in Scenario S1 than in Scenario S2, with values of 5.43 (SD = 0.95), 4.28 (SD = 1.09), and 2.72 (SD = 1.12), respectively, for Scenario S1 and 5.72 (SD = 0.62), 4.50 (SD = 0.96), and 2.97 (SD = 1.06), respectively, for Scenario S2. 

Module III addressed Positive and Negative emotions. Positive emotions in Scenario S1 were 3.69 (SD = 1.16) and in Scenario S2 they were 3.76 (SD = 1.20). Negative emotions in Scenario S1 were 2.60 (SD = 1.06), and in Scenario S2 they were 2.38 (SD = 0.94). We observe that Positive emotions are higher and Negative emotions are lower in Scenario S2 than in Scenario S1. 

The Intention to use and Product loyalty scales of Module IV were higher in Scenario S2 than in Scenario S1. In Scenario S1, the Intention to use scale was 4.27 (SD = 1.40), and the Product loyalty scale was 4.05 (SD = 1.15). In Scenario S2, the Intention to use scale was 4.68 (SD = 1.16), and the Product loyalty scale was 4.38 (SD = 0.75). 

Module V addressed Overall evaluation. The Overall evaluation scale in Scenario S1 scored 3.1, SD = 1.6. The Overall evaluation scale in Scenario S2 scored 3.5, SD = 1.2. The scale scores indicate that the scenario using the DARS Traffic Plus mobile application while driving received a higher score. 

A check for normal distribution of the meCUE 2.0 questionnaire data showed that the data were not normally distributed. Therefore, we used the Wilcoxon signed-rank test. We detected a significant improvement in the scales of Scenario S2, driving with the DT+ mobile application, for the Usefulness (Z = −4.26, *p* = 0.001), Visual aesthetics, Commitment (Z = −5.02, *p* = 0.001), Negative emotions (Z = −4.52, *p* = 0.001), Intention to use (Z = −3.27, *p* = 0.001), and Overall evaluation (Z = −4.05, *p* = 0.001) scales. The results are presented in Table 3.

### 4.4. Driving Simulator Data Insights

The quantitative driving parameters were recorded while driving in a driving simulator, in addition to eye-tracking data. With quantitative driving parameters, we describe the average number of driving events, such as fast steering movement, hard braking, driving too slow or too fast, and similar, in Table 4. The table presents the quantitative driving parameters data for Scenario S1 (without DT+ mobile application) and Scenario S2 (with DT+ mobile application).

During the driving scenarios, we observed whether participants were driving at the speed limit, below the speed limit, or above the speed limit; however, due to a driving simulator limitation, we were not able to detect the actual speed at which they were driving. We set the 5 km/h interval from the current speed limit to determine how they drive. If they were driving more than 5 km/m above the speed limit, they triggered the “Driving too fast” event. If they were driving more than 5 km/m below the speed limit, they triggered the “Driving too slow” event. The driving simulator parameter “Too fast acceleration” was triggered if participants pressed the accelerator pedal to the maximum power position for more than 5 s, and similar for the driving parameter “Hard braking”. 

In Scenario S1, we detected the quantitative driving parameter “Too fast acceleration” as the most frequent, with an average of 2416.52 events (SD = 1450.7) recorded. The second most frequent quantitative driving parameter event was “Too small safety distance”, with 634.85 (SD = 319.13) events recorded. We detected some instances of “Driving too slow”, with 15.88 (SD = 11.92) events recorded, and “Driving too fast”, with 27.97 (SD = 26.2) events recorded. On average, drivers viewed the “Traffic sign event” during driving 47.7 (SD = 17.97) times. There were some small contacts between cars while driving, an average of 1.67 (SD = 3.57) contacts per scenario, with only one severe car accident.

In Scenario S2, “Too fast acceleration” was the most frequent quantitative driving parameter, with an average of 4396.94 (SD = 1774) events recorded. The second most frequent driving performance event was “Too small safety distance”, with 1134.58 (SD = 180.32) events. We detected some instances of “Driving too slow”, with 21.52 (SD = 27.67) events, and “Driving too fast”, with 58.61 (SD = 49.70) events. While driving Scenario S2 with the DT+ mobile application, participants viewed traffic signs more frequently, with 117.42 (SD = 18.80) events. There were some small contacts between cars in Scenario S2 while driving, on average 1.88 (SD = 3.98), with no severe car accidents. In addition, we noticed an increase in the “Too small safety distance” quantitative driving parameter, with 1134.58 (SD = 180.32) events in Scenario S2 compared to 634.85 (SD = 319.13) events in Scenario S1.

We checked the quantitative driving parameter data for the normal distribution of the data. The data were not normally distributed. Therefore, we used the Wilcoxon signed-rank test. There are significant differences in the parameters “Fast steering movement” (Z = −2.05, *p* = 0.040), “Hard braking” (Z = −3.21, *p* = 0.001), and “Driving too slow” (Z = −2.58, *p* = 0.010). In Scenario S2, the parameters listed previously had lower values than in Scenario S1. Results show that the participants drove with fewer fast steering movements, did less hard braking, and drove below the speed limit. The data are presented in Table 5. 

We analysed the quantitative driving parameters in even greater detail; an analysis was performed for all listed quantitative driving parameter intervals where Hazardous Locations Notification events were present in the scenarios. The first analysis was conducted on the frequency of listed quantitative driving parameters in the locations of Hazardous Locations Notification events for Scenario S1 (without DT+ mobile application) and Scenario S2 (with DT+ mobile application) in the driving simulator (Table 6). The second analysis was conducted for the time duration in seconds of listed quantitative driving parameters in the locations of Hazardous Locations Notification events for Scenario S1 (without DT+ mobile application) and Scenario S2 (with DT+ mobile application) in the driving simulator (Table 6). The data were found to be not normally distributed and had a nonparametric distribution. We used the Wilcoxon signed-rank test.

We found a significant difference only in the quantitative parameter of “Driving too slow” (Z = −2.97, *p* = 0.003). The mean value of the event was reduced from 2701.64 events (SD = 826.59 events) to 2462.15 events (SD = 1064.22 events) in Scenario S2 when drivers used the DARS Traffic Plus application. Additionally, we detected a significant reduction in time spent in Hazardous Locations Notification traffic event intervals (Z = −3.69, *p* = 0.001) for the quantitative driving parameter of “Driving too slow”.

### 4.5. Reporting and Receiving Hazardous Locations Notification Events in the DT+ Mobile Application

The DT+ mobile application informed participants while driving about Hazardous Locations Notification events with a series of on-screen messages. Participants also had the opportunity to report Hazardous Locations Notification events. We measured the interaction with the DT+ mobile application in Scenario S2, as well as driving with the DT+ mobile application in Scenario S3, where we tested the extended functionalities of the DT+ mobile application.

On average, as presented in Table 7a,b, participants in Scenario S2 saw 4.97 messages (SD = 1.69) out of the 7 messages shown in the driving scenario and sent 2.97 messages (SD = 1.63 messages) of the 6 expected messages. In Scenario S3, the drivers saw 9.06 messages (SD = 0.35) out of the 11 shown in the driving scenario and sent 1.94 messages (SD = 1.6) of the 5 expected messages. For the number of pressed buttons, there was not a set upper limit for this parameter. 

### 4.6. Time Distributions of Viewing of Areas of Interest (AOIs)

The driving simulator had six predefined areas of interest (AOIs): middle screen, left screen, right screen, dashboard, dashboard GSM, and dashboard speedometer. Areas of interest are presented in Figure 4. 

Participants’ average time to complete Scenario S1 was 520.34 s, and 522.14 s in Scenario S2. We did not find a significant difference in the overall time for completion of Scenario S1 and Scenario S2. The time distribution over the areas of interest for Scenarios S1 and S2 was similar. The main difference we detected in the amount of time observed by participants was the area of interest road directly ahead, with 456.83 s (SD = 51.12 s) in Scenario S1. In Scenario S2, the time spent looking directly ahead on the road decreased to 434.62 s (SD = 53.63 s). The difference in time represents 5.42% of the total average mean time needed to complete the scenario and presents the time drivers focused on the DT+ mobile application (23.57 s, SD = 14.7 s). The time for other areas of interest (left and right mirrors, dashboard, and speedometer) did not change significantly. In Scenario S2, the participants were looking at the left screen area of interest (meaning left mirror) for 5.77 s (SD = 3.98 s), the right screen area of interest (meaning right mirror) for 19.51 s (SD = 16.56 s), the dashboard area of interest for 20.03 s (SD = 12.20 s), and the dashboard speedometer area of interest for 18.64 s (SD = 8.85 s). The distribution of time viewed in the areas of interest in Scenario S2 is presented in Figure 5, and the average time viewed for the areas of interest for Scenario S1 (without DT+ mobile application) and Scenario S2 (with DT+ mobile application) in the driving simulator is shown in Table 8.

In Figure 6, we examined the maximum continuous time for participants viewing the area of interest of the dashboard GSM. This time represents the maximum time that participants viewed the mobile phone screen without looking away from the mobile phone. The average time of continuous looking at the mobile phone screen was 1.64 s. Seven participants had the most extended view of the dashboard GSM area of interest over a 2-s duration. Except for one participant, for all the other participants, the second most extended view of the dashboard GSM area of interest was below two seconds. The 2-s threshold is of high significance for driving. The 2-s threshold was set by the traffic regulations for safety distance and, in some cases, a time for takeover action in the case of an emergency. The 2-s threshold is presented with a red line in Figure 6.

### 4.7. Eye Tracking

In the driving simulator, we collected a heatmap and scan path data in addition to quantitative driving parameters. We counted as one gaze the user focus of 20 ms of uninterrupted gaze at the same spot inside the specific area of interest. At the same time, gaze coordinates (position x and position y) were recorded. The origin coordinate (0,0) point of the area of interest was always in the top left corner. The three main 50-inch screens had 4K resolution; their size was 111.76 cm by 64.44 cm, while the dashboard size was 90 cm by 45 cm, the dashboard speedometer had a size of 35 cm by 14 cm, and the dashboard GSM had a size of 16 cm by 21 cm. In addition to the screen position of the gaze, we collected the number of fixations during a specific drive, the start time, and the fixation duration. The heatmap and scan path data in Figure 7 and Figure 8 represent the 3-s interval before and the 2-s interval after receiving the Hazardous Locations Notification event on the road. An example of the heatmap data is shown in Figure 7, where the area of greater focus is coloured with higher intensity and warmer colours. The scan path data show saccadic eye movements while viewing and recognising patterns, as shown in Figure 8.

### 4.8. Post-Experiment Interview

After the driving scenarios, we asked the participants in the pilot evaluation study to rate the DT+ mobile application functionalities using the five-point Likert scale. A higher value on the five-point Likert scale represents a more favourable score. Among the DT+ mobile application functionalities we asked participants to rate were the mobile application’s home screen, the colour palette used, the clarity of the icons, the way notifications were sent, the usability, and the user experience of the DT+ mobile application. 

Participants selected the most suitable view for the first screen of the application as a view with a map (75.8%) rather than a view with icons representing functionalities in the application (24.2%), as shown in Table 9.

Table 10 presents participants’ feedback on additional DT+ mobile application functionalities. Participants reported positive clarity regarding the colour palette used for day and night views. The day view colour palette received a score of 4.61 (SD = 0.56) and the night view colour palette received a score of 4.00 (SD = 0.97). The functionality of the built-in zoom functions in the application received a mark of 4.67 (SD = 0.59). The clarity of icons at 4.64 (SD = 0.60) and the clustering of icons at 4.82 (SD = 0.46) were also marked positively during the screen view magnification. The size of the interaction field around buttons in the application was found to be sufficient, 4.55 (SD = 0.83). The DT+ mobile application received a high mark for a perceived usability value of 4.79 (SD = 0.48) and a perceived user experience of 4.58 (SD = 0.71). The drivers have indicated a high possibility of future application use with a score of 3.91 (SD = 1.01).

Reporting traffic events and receiving a Hazardous Locations Notification for traffic events in the DT+ mobile application are some of the most important functionalities. Participants were informed via Hazardous Locations Notification of the upcoming hazardous location with a series of on-screen notifications on the DT+ mobile application. The furthest away notification from the hazardous location had a blue background colour with an icon of one of four available Hazardous Locations Notification events, a short description of the notification, and the distance to the hazardous location. This notification was shown from 2.5 to 3 km in advance of the hazardous locations on the road. Approximately 1 km from the hazardous location, an orange notification was shown with the same visual information as the previous notification. The notification turned red around 300 to 400 m from the hazardous location, prompting immediate action. The perceived level of safety for drivers was higher, 4.30 (SD = 0.77), for those receiving safety Hazardous Locations Notifications while driving. When sending Hazardous Locations Notifications of traffic events while driving, drivers’ perceived safety level was lower, 3.64 (SD = 1.08). Participants wished to receive the Hazardous Locations Notifications for traffic events from the top side of the mobile phone screen, 3.82 (SD = 1.13), rather than from the bottom side of the mobile phone screen, 3.55 (SD = 1.20). Recommended audio notifications accompanying visual Hazardous Locations Notifications are voice notifications at 4.55 (SD = 0.71) and notifications with a beep at 4.33 (SD = 0.85), followed by notifications without a sound or beep at 3.82 (SD = 1.13). The results are shown in Table 11.

When reporting traffic events on the road, the most relevant Hazardous Locations Notification event was identified as Reporting Poor Visibility, followed by Obstacle on the Road, Traffic Jam Ahead, and Accident Zone. The order of relevance remained unchanged when receiving a Hazardous Locations Notification for traffic events, with the most critical notification being Reporting Poor Visibility, followed by Obstacle on the Road, Traffic Jam Ahead, and Accident Zone. We asked the participants to rank the Hazardous Locations Notifications by their perceived importance to them. Which notifications were the most important to them, and would they like to receive them all the time? An example of a Hazardous Locations Notification received by participants within the DT+ mobile application is shown in Figure 9. Similarly, we asked the participants to rank the Hazardous Locations Notifications for the case when they reported the events. Which hazardous location event would they be most likely to report? We used the five-point Likert scale, where 1 represents low interest in the Hazardous Locations Notifications type and 5 represents high interest in the Hazardous Locations Notifications type. When reporting Hazardous Locations Notifications, the participant first pressed the big icon for reporting located on the bottom of the left side of the mobile phone screen. A screen with all four Hazardous Locations Notifications icons was presented. Below the Hazardous Locations Notifications icons, two arrows indicated if the hazardous location was in the direction of travel or the opposite direction. Participants had to press the “Report” button to complete the reporting. The results are listed below in Table 12. 

## 5. Discussion

The study participants were regular, healthy drivers who drove daily on the roads, and they primarily had a dynamic driving style and obeyed the road speed limitations. Most of them use mobile smartphones and applications, and at least one mobile application is used for driving assistance. They primarily drive alone in the car and use the cruise control system as their primary safety assist system. 

Participants treat daily migrations and one-time non-daily migrations differently. With daily migrations, participants check for driving conditions in the car or while driving. For the one-time non-daily migrations, participants check for driving conditions before getting into the car. The usage of mobile applications for checking traffic conditions on the roads is comparable in both types of migrations. 

Q1: Does the DARS Traffic Plus mobile application provide a good user experience for the driver?

Overall, the assessment of the DARS Traffic Plus mobile application was very positive. Drivers rated the DARS Traffic Plus mobile application positively in the interview after the driving scenarios and the user experience questionnaire. 

We designed the driving simulator protocol in line with the recommendations of Ref. [22] for the adaptation period before the research driving session. The road used for the adaptation was not demanding. As suggested in the literature, an approximate 7-min driving time was in line with recommendations [22]. 

The day view colour palette was marked higher than the night view colour palette, as most drivers use the day view setting on their mobile smartphones. Participants determined that the preferred position for showing notifications was from the top of the mobile smartphone. The DARS Traffic Plus mobile application follows the human–machine interface recommendations as suggested by Ref. [17]. The HMI elements are grouped together in the application; the visual interface has efficient colour contrast between foreground and background; and the messages are short and clear. The voice was marked as the best notification sound, followed by the beep sound. The interview highlighted that the voice notification should be reserved for the most critical traffic notifications, such as an emergency vehicle on the road or driving in the wrong direction. Everyday use of the voice as a sound notification would lower its priority, and people would pay less attention to it. Therefore, participants found the use of the beep as an audio notification to be adequate. Our conclusions confirm the findings of Ref. [11], who suggested that collision warning and voice control applications are the most accepted applications for road safety. The map view received a higher score than the icon view of the application’s first screen and is more commonly used in mobile applications for driving. No significant complaints about the icons’ design, size, or clustering were reported. 

We measured the user experience of the DARS Traffic Plus mobile application with two tools. The first tool was the subjective measurement of the user experience in the post-experiment interview. The second tool was the meCUE 2.0 questionnaire. All the tools used indicate a good value for the user experience. Participants reported an excellent overall user experience in the post-experiment interview. Additionally, the DARS Traffic Plus mobile application provides usability for the users. 

The overall evaluation scale of the meCUE 2.0 user experience questionnaire was positive in all scenarios, and the highest score was achieved in Scenario S3. We noticed a similar trend for other scales, except for the scale of Negative emotions. The most heightened reported negative emotions were in Scenario S1, where users drove without the DARS Traffic Plus mobile application. The Negative emotions scale lowered in Scenario S2 and reached the lowest scale value in Scenario S3. In Scenario S3, participants interacted the most with the mobile application and tested various application features. We can conclude that using the DARS Traffic Plus mobile application does not significantly increase negative emotions or irritate users. The scores for the scales of Usefulness and Usability received high marks and supported the responses to the user experience and usability questions from the post-experiment interview. Similarly, the Visual aesthetics, Intention to use, and Product loyalty scales received the highest marks in Scenario 3, which again supports the post-experiment interview responses.

Q2: Does the DARS Traffic Plus mobile application impair the driving performance of the driver?

The eye tracking system had six areas of interest defined: left screen, middle screen, right screen, dashboard, speedometer, and dashboard GSM (mobile phone). We have not noticed significant differences in the participants’ average times needed to complete Scenario S1 and Scenario S2. The mean value of the average time difference to complete Scenarios S1 and S2 is less than 2 s. The main difference in time distribution recorded over the areas of interest for a particular scenario was the time drivers were focused directly on the road ahead (i.e., middle screen). In Scenario S2, the summary of combined times when drivers were focused directly on the road ahead and the time they were focused on the mobile phone is almost identical to when the drivers were focused directly on the road ahead (i.e., middle screen). The time of focus on the other areas of interest did not change between the scenarios. 

The average mean value of the maximum duration of mobile phone views was 1.64 s. The longest view of the mobile phone was 3.49 s. Only 7 participants out of 33 had the most extended view of the mobile phone longer than 2 s; all the other participants had the longest view of the mobile phone shorter than 2 s. The average mean value of the mobile phone view is under 2 s [19], which is recommended as the total time to find the relevant information within a display. Additionally, the average mean value of the duration of the mobile phone view is under the time for a takeover request (TOR), as recommended by Ref. [19]. 

Participants were focused on driving and not on their mobile phones for a prolonged period of time. The average mean value of the maximum duration of the mobile phone was 1.64 s, and the twenty-six participants’ most extended view of the mobile phone was less than 2 s. The results are under the recommended safety limit for safety distance and the recommended time for a takeover request (TOR) [19]. Measured data indicate that interaction without severe adverse effects with the DARS Traffic Plus mobile application can be performed in cars while driving. It is important to note that the mobile phone must be placed securely in the phone holder. Participants only interact with the DARS Traffic Plus mobile application to receive and send hazardous location notifications. We conclude that the DARS Traffic Plus mobile application does not impair the driver’s driving performance.

Q3: Does the DARS Traffic Plus mobile application provide measurable and positive differences in driving parameters when driving?

Additionally, we analysed only intervals where Hazardous Locations Notification traffic events occurred in Scenarios S1 and S2. We detected a change in one simulator event parameter, and this parameter was “Driving too slow”. The mean value of the parameter was reduced from Scenario S1 to Scenario S2. A similar study [21] detected a faster speed increase when drivers were aware of road condition changes in advance. The test showed significant differences in time spent in Hazardous Locations Notification traffic event intervals. Drivers spent less time in Hazardous Locations Notification traffic event intervals in Scenario S2. Results are shown in Table 4, Table 5 and Table 6. 

One possible explanation for the reduction in driving time and the simulator event parameter “Driving too slow”, is that the drivers adjusted their driving style ahead of the Hazardous Locations Notification traffic event. Drivers were informed about traffic events in advance through the Hazardous Locations Notification in the DARS Traffic Plus application. They adjusted their driving style and, in advance, re-evaluated conditions on the road. When drivers were confronted with the same traffic events without advanced notifications, they had to adjust their driving style at the point of the traffic event location. When driving with the DARS Traffic Plus application, we observed that drivers were driving less erratically and passed through Hazardous Locations Notification traffic events faster. 

We conclude that the DARS Traffic Plus mobile application provides a measurable and positive difference in driving parameters when driving. We detected a significant difference in the driving parameter “Driving too slow”. The reduction in driving time combined with the driving parameter “Driving too slow”, suggests that participants could adjust their driving based on the received Hazardous Locations Notification. They were informed in advance and were able to adjust their driving accordingly to the conditions on the road. 

Q4: Does the DARS Traffic Plus (DT+) mobile application increase the feeling of safety while driving?

An essential aspect of the pilot study was the evaluation of the perceived safety of the functionalities for reporting and receiving the Hazardous Locations Notification. Participants reported a high perceived value of safety for receiving the Hazardous Locations Notification through the DARS Traffic Plus mobile application. The perceived value of safety was lower when participants were reporting a Hazardous Locations Notification on the road. The primary reported reasons for lower perceived safety value were interaction with the mobile phone while driving and looking away from the road. Reporting or receiving the C-ITS notifications did not impact the perceived value and order of importance of the four presented Hazardous Locations Notifications for participants. We conclude that the DARS Traffic Plus mobile application increases the feeling of safety while driving. 

### 5.1. Limitations and Mitigations

The number of participants could be increased in future evaluations to provide additional validation and insights. However, the 33 participants represent a statistically significant group, and the results gathered through the pilot study evaluation present meaningful insights into users’ driving behaviour using supporting mobile applications. As the time of evaluation for a single participant was, on average, approximately 90 min, the higher number of participants would cause many time constraints for the execution of the evaluation. 

The professional driving simulator was the most prominent centrepiece of the evaluation, combined with the DARS Traffic Plus application. The experience of driving in the driving simulator had some impact on the participants, but the effects diminished with every additional drive in the simulator. An improved driving simulator with additional sense feedback during driving could improve the evaluation outcome.

### 5.2. Conclusions and Future Work

In this publication, we provided the results of the user experience evaluation of the DARS Traffic Plus mobile application. The research shows that drivers in the evaluation study showed positive opinions about using the DARS Traffic Plus mobile application while driving. They felt more informed while driving with the mobile application and had a higher perceived level of safety. The use of the DARS Traffic Plus mobile application did not significantly impact the drivers’ driving performance in the evaluation. Furthermore, the pilot study evaluation in the driving simulator provided deep insights into driving with the DARS Traffic Plus mobile application. Participants reported an elevated level of safety when receiving the Hazardous Locations Notifications warning them about possible hazardous events on the road. The perceived value of safety was lower when participants reported Hazardous Locations Notifications on the road. However, the level of safety was higher than in the case of driving without the DARS Traffic Plus mobile application. In the interviews, participants marked the graphic design and use of colours in the application as good. Overall, the DARS Traffic Plus mobile application provides drivers with a good user experience and usability. 

The pilot study evaluation of the DARS Traffic Plus mobile application in the driving simulator should continue. Participants provided good insights into the possible improvements to DARS Traffic Plus mobile application functionalities. Additional research efforts should be directed towards the interaction of the driver with the assistive mobile application and the understandability of the safety notifications received. Improvements could also be made to the driving simulator. 

We want to thank DARS d.d. company for financing the driving simulator’s DARS Traffic Plus mobile application evaluation. The authors acknowledge the financial support from the Slovenian Research Agency (research core funding No. P2-0450 and No. P2-0425).

## Figures and Tables

**Figure 1 sensors-24-04948-f001:**
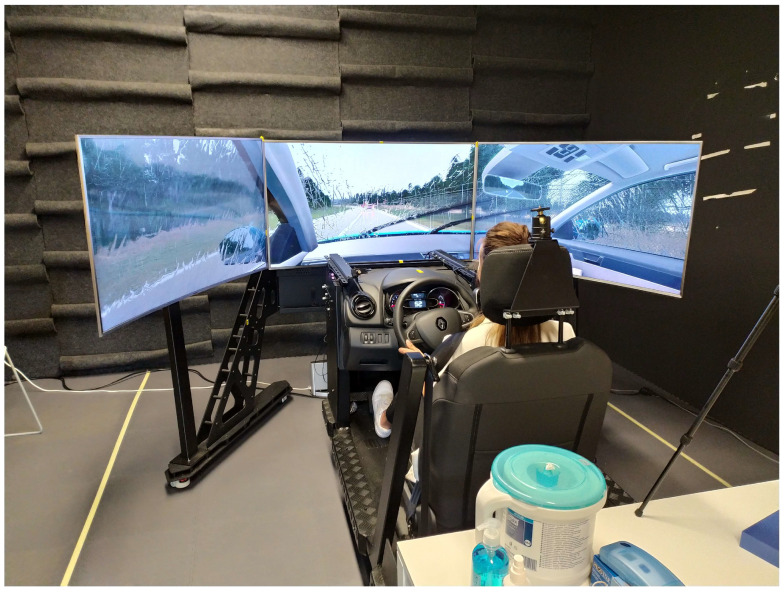
A professional driving simulator the DARS company used to assess the existing highway road network efficiency and future development of the road network and to evaluate the DARS Traffic Plus mobile application.

**Figure 2 sensors-24-04948-f002:**
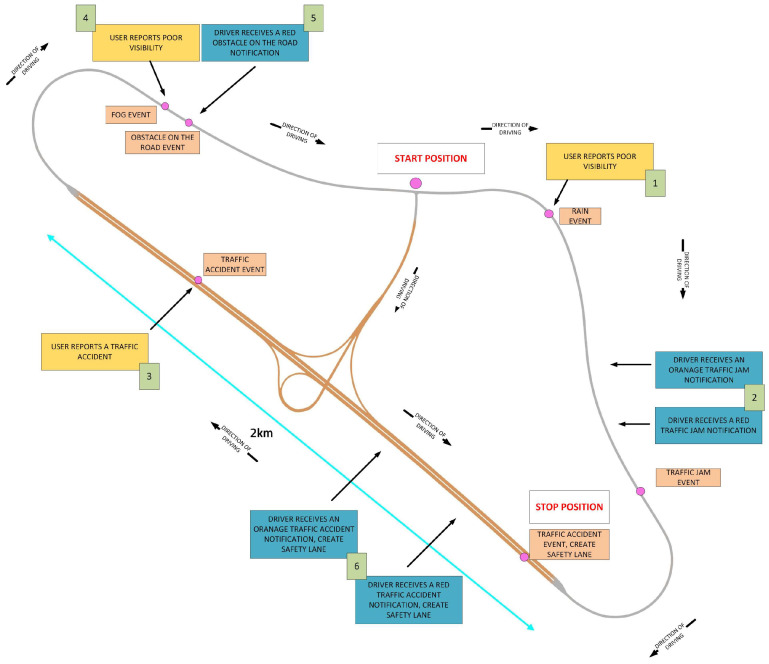
Scenario S2 diagram with the scenario’s marked start and end positions and locations of Hazardous Locations Notification warning events.

**Figure 3 sensors-24-04948-f003:**
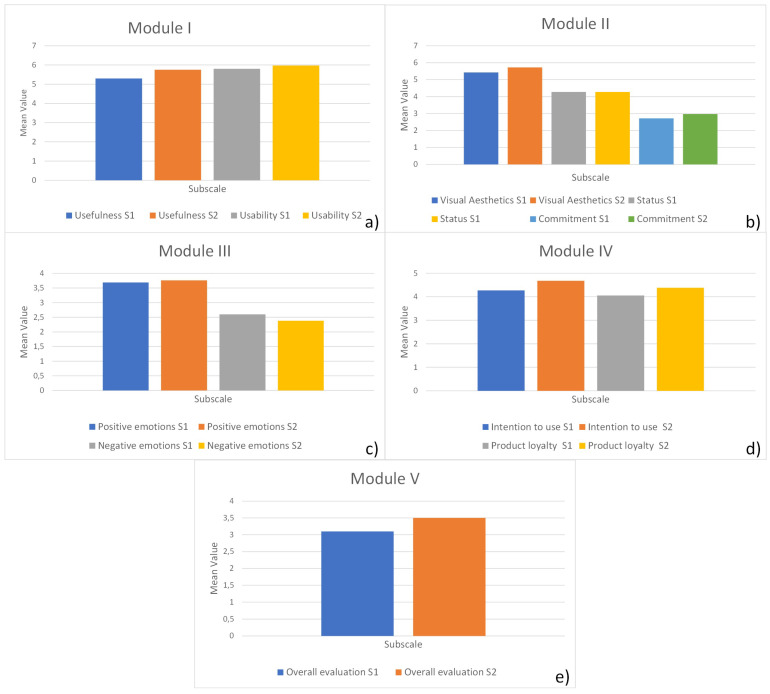
Graphs of meCUE 2.0 questionnaire modular subscales for two driving scenarios, Scenario S1 (without DT+ mobile application) and Scenario S2 (with DT+ mobile application), in the driving simulator. (**a**) Module I (Usefulness and Usability scales); (**b**) Module II (Visual aesthetic, Status, and Commitment scales); (**c**) Module III (Positive and Negative emotions scales); (**d**) Module IV (Intention to use and Product loyalty scales); (**e**) Module V (Overall evaluation scale).

**Figure 4 sensors-24-04948-f004:**
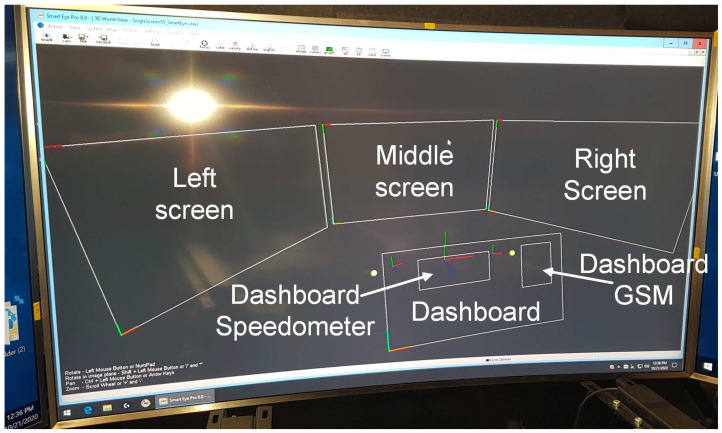
Visual representation of six predefined areas of interest (AOIs).

**Figure 5 sensors-24-04948-f005:**
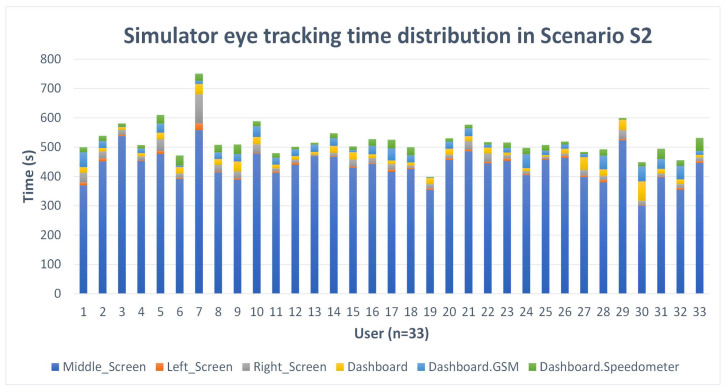
Simulator eye tracking time distribution for various areas of interest in Scenario S2.

**Figure 6 sensors-24-04948-f006:**
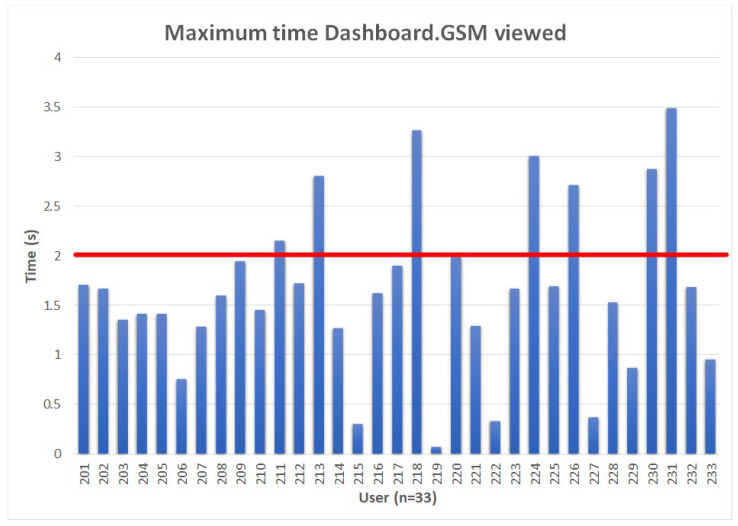
Maximum continuous time viewing the dashboard GSM area of interest in seconds. The red bar represents the prescribed safety distance of two seconds [16].

**Figure 7 sensors-24-04948-f007:**
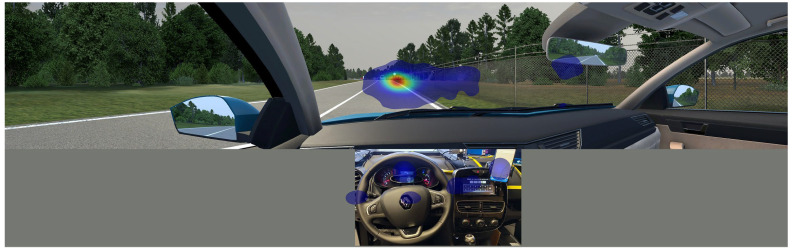
Eye tracking heatmap data. The area of greater focus is coloured with higher intensity and warmer colours.

**Figure 8 sensors-24-04948-f008:**
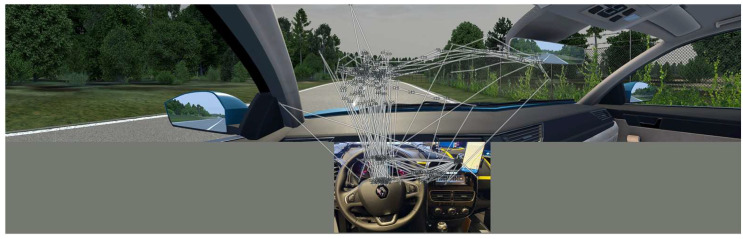
Eye tracking scan path data.

**Figure 9 sensors-24-04948-f009:**
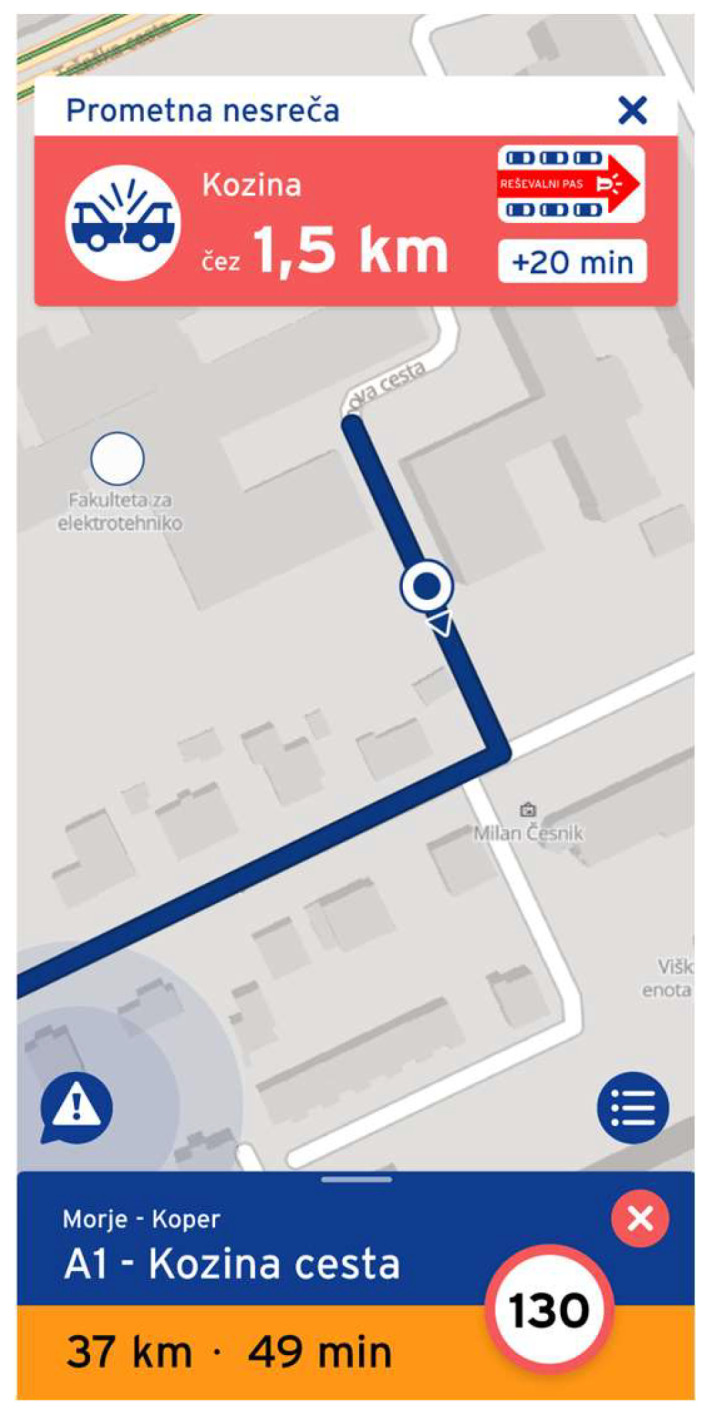
An example of a Hazardous Locations Notification received within the DT+ mobile application. The user interface is in Slovenian, as the study was conducted in Slovenia.

**Table 1 sensors-24-04948-t001:** Description of scenarios used to evaluate the DARS Traffic Plus mobile application.

Scenario Name:	Scenario Description:
Scenario S0	Scenario S0 was the introduction scenario, where participants became familiar with the driving simulator. Scenario S0 included driving on motorways, main roads, and local roads with limited traffic.
Scenario S1	Scenario S1 included driving on motorways, main roads, and local roads with variable traffic on the route, stationary vehicles, vehicle accidents, and severe weather conditions.
Scenario S2	Scenario S2 was similar to Scenario 1, including the same Hazardous Locations Notification events. Scenario S2 also included driving on motorways, main roads, and local roads with variable traffic on the route, stationary vehicles, vehicle accidents, and severe weather conditions with the help of the DARS Traffic Plus mobile application.
Scenario S3	Scenario 3 was intended to test additional DT+ mobile application functionalities. Participants could report or receive messages through the DT+ mobile application for four selected Hazardous Locations Notification events. The first additional task was to alter the pre-selected navigation route in the DT+ mobile application. The second task was checking and counting the Hazardous Locations Notification events on the new route. We instructed participants to comply with the received DT+ mobile application messages.

**Table 2 sensors-24-04948-t002:** Comparison of meCUE 2.0 questionnaire module subscale results for Scenario S1 (without DT+ mobile application) and Scenario S2 (with DT+ mobile application). Important higher values of Scenario S2 subscales are highlighted in bold text.

		Scenario S1 (without DT+ Mobile Application)			Scenario S2 (with DT+ Mobile Application)		
Module	Subscale	**Average Mean Value**	**S.D. Value**	**Min/Max Subscale Value**	**Average Mean Value**	**S.D. Value**	**Min/Max Subscale Value**
Module I	Usefulness	5.30	1.02	3.67/7.00	**5.75**	**0.63**	4.67/7.00
	Usability	5.80	0.85	3.67/7.00	5.97	0.75	4.67/7.00

Module II	Visual aesthetics	5.43	0.95	3.67/7.00	**5.72**	**0.62**	4.67/7.00
	Status	4.28	1.09	1.67/6.00	4.50	0.96	2.00/6.00
	Commitment	2.72	1.12	1.00/4.67	2.97	1.06	1.00/4.33

Module III	Positive emotions	3.69	1.16	1.00/5.67	3.76	1.20	1.33/6.00
	Negative emotions	2.60	1.06	1.00/5.00	**2.38**	**0.94**	1.00/4.50

Module IV	Intention to use	4.27	1.40	1.67/7.00	**4.68**	**1.16**	2.00/6.67
	Product loyalty	4.05	1.15	2.33/6.00	4.38	0.75	3.33/6.00

Module V	Overall evaluation	3.1	1.6	−0.5/4.5	**3.5**	**1.2**	−0.5/4.5

**Table 3 sensors-24-04948-t003:** Wilcoxon signed-rank test results for significant differences between meCUE 2.0 questionnaire subscales for Scenario S1 (without DT+ mobile application) and Scenario S2 (with DT+ mobile application).

Quantitative Driving Parameters	Z	Asymp. Sig. (2-Tailed)
Usefulness_S1–Usefulness_S2	**−4.26**	**0.001**
Usability_S1–Usability_S2	−0.64	0.522
Visual_aesthetics_S1–Visual_aesthetics_S2	**−3.15**	**0.002**
Status_S1–Status_S2	−0.74	0.455
Commitment_S1–Commitment_S2	**−5.02**	**0.001**
Positive_emotions_S1–Positive_emotions_S2	−1.53	0.126
Negative_emotions_S1–Negative_emotions_S2	**−4.52**	**0.001**
Intention_to_use_S1–Intention_to_use_S2	**−3.27**	**0.001**
Product_loyalty_S1–Product_loyalty_S2	−2.21	0.027
Overall_evaluation_S1–Overall_evaluation_S2	**−4.05**	**0.001**

**Table 4 sensors-24-04948-t004:** Comparison of quantitative driving parameters in the driving simulator for Scenario S1 (without DT+ mobile application) and Scenario S2 (with DT+ mobile application).

Driving Events:	Average Number of Events for Drivers in Scenario S1 (without DT+ Mobile Application)		Average Number of Events for Drivers in Scenario S2(with DT+ Mobile Application)	
	Average Mean Value	S.D. Value	Average Mean Value	S.D. Value
Wrong way of driving	3.67	3.56	5.82	2.99
Driving reverse	0.58	0.83	0.45	1.86
**Fast steering movement**	**7.21**	**6.93**	**1.39**	**3.06**
**Hard braking**	**8.24**	**11.25**	**11.25**	**0.18**
**Driving too slow**	**15.88**	**11.92**	**21.52**	**27.67**
Driving too fast	27.97	26.2	58.61	49.70
Driving outside driving lane	8.91	4.97	21.76	9.27
Too small safety distance	634.85	319.13	1134.58	180.32
Accelerating too fast	2416.52	1450.7	4396.94	1774.33
Traffic sign event	47.7	17.97	117.42	18.80
Car accident (broken windshield)	0.03	0.17	0.00	0.00
New car start	0.03	0.17	0.00	0.00
Car accident (small incident)	1.67	3.57	1.88	3.98

**Table 5 sensors-24-04948-t005:** Results for significant differences in quantitative driving parameters for Scenario S1 (without DT+ mobile application) and Scenario S2 (with DT+ mobile application) in the driving simulator.

Quantitative Driving Parameters	Z	Asymp. Sig. (2-Tailed)
Wrong way of driving S1–Wrong way of driving S2	−1.46	0.144
Driving reverse S1–Driving reverse S2	−0.74	0.461
**Fast steering movement S1–Fast steering movement S2**	**−2.05**	**0.040**
**Hard braking S1–Hard braking S2**	**−3.21**	**0.001**
**Driving too slow S1–Driving too slow S2**	**−2.58**	**0.010**
Driving too fast S1–Driving too fast S2	−1.15	0.250
Driving outside driving lane S1–Driving outside driving lane S2	−1.47	0.140
Too small safety distance S1–Too small safety distance S2	−0.37	0.708
Accelerating too fast S1–Accelerating too fast S2	−0.22	0.823
Traffic sign event S1–Traffic sign event S2	−0.62	0.530
Car accident (broken windshield) S1–Car accident (broken windshield) S2	−1.00	0.317
New car start S1–New car start S2	−1.34	0.180
Car accident (small incident) S1- Car accident (small incident) S2	−0.92	0.358

**Table 6 sensors-24-04948-t006:** Frequency (a) and time (b) significance analysis of quantitative driving parameters for Scenario S1 (without DT+ mobile application) and Scenario S2 (with DT+ mobile application) in the driving simulator.

	(a) Frequency Significance Analysis of Quantitative Driving Parameters for Scenario S1–Scenario S2		(b) Time Significance Analysis of Quantitative Driving Parameters for Scenario S1–Scenario S2	
**Quantitative Driving Parameters**	**Z**	**Asymp. Sig. (2-Tailed)**	**Z**	**Asymp. Sig. (2-Tailed)**
Wrong way of driving S1–Wrong way of driving S2	−0.61	0.541	−1.76	0.079
Driving reverse S1–Driving reverse S2	−0.27	0.785	0.00	1.000
Fast steering movement S1–Fast steering movement S2	−0.81	0.418	−1.01	0.309
Hard braking S1–Hard braking S2	−1.56	0.120	−1.20	0.229
**Driving too slow S1–Driving too slow S2**	**−2.99**	**0.003**	**−4.11**	**0.001**
Driving too fast S1–Driving too fast S2	−1.56	0.120	−0.06	0.950
Driving outside driving lane S1–Driving outside driving lane S2	−0.58	0.561	−1.81	0.070
Too small safety distance S1–Too small safety distance S2	−0.24	0.809	−1.29	0.195
Accelerating too fast S1–Accelerating too fast S2	−0.95	0.344	−1.70	0.088
Traffic sign event S1–Traffic sign event S2	−1.62	0.106	−0.77	0.437
Car accident (broken windshield) S1–Car accident (broken windshield) S2	−1.00	0.317	0.00	1.000
New car start S1–New car start S2	−1.00	0.317	0.00	1.000
Car accident (small incident) S1–Car accident (small incident) S2	−0.17	0.864	0.00	1.000

**Table 7 sensors-24-04948-t007:** Number of self-reported and received Hazardous Locations Notification traffic events in the DT+ mobile application.

	(a) Scenario S2 (with DT + Mobile Application)			(b) Scenario S3 (Extended Functionalities of DT + Mobile Application)		
	Mean Average Value	S.D. Value	Maximum Number of Events	Mean Average Value	S.D. Value	Maximum Number of Events
Number of sent messages	2.97	1.63	6	1.94	1.60	5
Number of pressed buttons	3.42	1.70	--	7.82	3.70	--
Number of viewed messages	4.97	1.69	7	9.06	0.35	11

**Table 8 sensors-24-04948-t008:** Distribution of average time viewed for areas of interest for Scenario S1 (without DT+ mobile application) and Scenario S2 (with DT+ mobile application) in the driving simulator.

Area of Interest (AOI)	Average Time AOI Scenario S1 (without DT+ Mobile Application) (s)	S.D. Value (s)	Average Time AOI Scenario S2 (with DT+ Mobile Application) (s)	S.D. Value (s)
Saso_Middle_Screen	456.84	51.12	434.61	53.64
Saso_Left_Screen	5.19	2.62	5.77	3.98
Saso_Right_Screen	20.44	11.45	19.51	16.57
Dashboard	17.38	14.04	20.03	12.20
Dashboard.GSM	-	-	23.57	14.71
Dashboard.Speedometer	20.49	11.05	18.65	9.85
Sum	520.34		522.14	

**Table 9 sensors-24-04948-t009:** Preferred home screen option in the DARS Traffic Plus mobile application as voted by participants in the pilot study evaluation.

Parameter	Value
First Screen View Maps	75.8%
First Screen View Icons	24.2%

**Table 10 sensors-24-04948-t010:** Results of the post-experiment interview in the pilot study evaluation of the DT+ mobile application functionalities using the five-point Likert scale.

Parameter	Average Mean Value	S.D. Value
Colour Palette for Day View in DT+	4.61	0.56
Colour Palette for Night View in DT+	4.00	0.97
Zoom Function	4.67	0.59
Clarity of Icons	4.64	0.60
Clustering of Icons	4.82	0.46
Size of Interaction Field	4.55	0.83
Usability Perceived	4.79	0.48
User Experience Perceived	4.58	0.71
Future DARS Traffic App Usage	3.91	1.01

**Table 11 sensors-24-04948-t011:** Results for the functionalities of reporting traffic events and receiving Hazardous Locations Notifications for traffic events for the DT+ mobile application.

Parameter	Average Mean Value	S.D. Value
Notifications from the Bottom of the Phone	3.55	1.20
Notifications from the Top of the Phone	3.82	1.13
Notifications Beep	4.33	0.85
Notifications No Beep	2.58	1.00
Notifications Voice	4.55	0.71
Safety Sending (Reporting) Traffic Events	3.64	1.08
Safety Receiving Notifications	4.30	0.77

**Table 12 sensors-24-04948-t012:** Rating of the importance of Hazardous Locations Notifications traffic events for reporting by the participants (upper part of the table) and receiving notification on the mobile phone screen (lower part of the table) for traffic events in the DT+ mobile app.

Participants Reporting Traffic Events	Average Mean Value	S.D. Value
Reporting Traffic Jam	2.45	1.03
Reporting Accident	1.48	0.87
Reporting Obstacles on the Road	2.64	0.90
Reporting Poor Visibility	3.42	0.75
**Participants Receiving Notifications for Traffic Events**	**Average Mean Value**	**S.D. Value**
Notifications Traffic Jam	2.39	1.12
Notifications Accident	1.55	0.87
Notifications of Obstacles on the Road	2.61	0.83
Notifications Poor Visibility	3.45	0.75

## Data Availability

The data presented in this study are available on request from the corresponding author.

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
