# Peer review of "Enhancing Driving Safety through User Experience Evaluation of the C-ITS Mobile Application: A Case Study of the DARS Traffic Plus App in a Driving Simulator Environment"

_sensors, 2024, doi:10.3390/s24154948_

Round 1

Reviewer 1 Report

Comments and Suggestions for Authors

The study presents an investigation to evaluate the effectiveness of the DARS Traffic Plus mobile application to improve driving safety conditions.  

The experimental design appears correct and tools and methods are described. However, the two types of analysis are not equally convincing.

 The first, based on a questionnaire survey, describes the results completely and comments on them correctly. The second, defined by the authors as quantitative analysis, presents many inaccuracies that require clarification.

On the other hand, the inadequate significance of objective investigations and measurements of indirect user actions clearly emerges from a sentence by the authors (line 128) combined with a non-existent bibliographical analysis in this regard.

It is therefore suggested, as a first action, to expand the bibliographic analysis by inserting studies relating to the analysis of risk perception or more generally safety conditions through measures of human driving behavior (kinematic-dynamic and/or psychophysiological measures). 

Concerning the effectiveness assessments through quantitative driving parameters, it is suggested to specify the reference numerical threshold values. It is necessary for the authors to clarify the meaning of "too fast acceleration" or "Driving too slow"... It is not even possible to consider it as a comparative evaluation since the quantities are highly non-homogeneous

Regarding the comparison of results between scenarios (tab.5, tab.6, tab.7) it is necessary to specify the evaluation scale adopted.

The meaning of the rating, the results of which are reported in table 12, is absolutely unclear. It is suggested to explain how the ratings can be made comparable.

Finally, in relation to the Time distributions of viewing of Area of interests, it is suggested to specify any correlations with the motion conditions, or other specific literature indicators to evaluate the levels of understanding and/or attention.

Author Response

Letter to Editor and Reviewers-3090723

Editor's and reviewers' comments analysis

Manuscript ID: sensors-3090723-R1

Type: Article

Title: Enhancing Driving Safety Through User Experience Evaluation of the C-ITS Mobile Application: A Case Study of the DARS Traffic Plus App in a Driving Simulator Environment

Authors: Gregor Burger, Jože Guna

This document contains the list of respected editor's and reviewers' comments on the manuscript " Enhancing Driving Safety Through User Experience Evaluation of the C-ITS Mobile Application: A Case Study of the DARS Traffic Plus App in a Driving Simulator Environment ", submitted to the Sensors Journal. We have found the comments very valuable. This document also contains the detailed authors' responses and explanations of changes done according to the editor's and reviewers' comments. When checking the line numbers in the article please use the "Show simple markup" Word view. All the reviewers' comments and suggestions were addressed.

Reviewers' Comments and Authors' Responses

Reviewer #1:

The study presents an investigation to evaluate the effectiveness of the DARS Traffic Plus mobile application to improve driving safety conditions.  

The experimental design appears correct and tools and methods are described. However, the two types of analysis are not equally convincing.

 The first, based on a questionnaire survey, describes the results completely and comments on them correctly. The second, defined by the authors as quantitative analysis, presents many inaccuracies that require clarification.

On the other hand, the inadequate significance of objective investigations and measurements of indirect user actions clearly emerges from a sentence by the authors (line 128) combined with a non-existent bibliographical analysis in this regard.

It is therefore suggested, as a first action, to expand the bibliographic analysis by inserting studies relating to the analysis of risk perception or more generally safety conditions through measures of human driving behavior (kinematic-dynamic and/or psychophysiological measures). 

Concerning the effectiveness assessments through quantitative driving parameters, it is suggested to specify the reference numerical threshold values. It is necessary for the authors to clarify the meaning of "too fast acceleration" or "Driving too slow"... It is not even possible to consider it as a comparative evaluation since the quantities are highly non-homogeneous

Regarding the comparison of results between scenarios (tab.5, tab.6, tab.7) it is necessary to specify the evaluation scale adopted.

The meaning of the rating, the results of which are reported in table 12, is absolutely unclear. It is suggested to explain how the ratings can be made comparable.

Finally, in relation to the Time distributions of viewing of area of interests, it is suggested to specify any correlations with the motion conditions, or other specific literature indicators to evaluate the levels of understanding and/or attention.

Reviewer #1:

Comment R.1.1              The study presents an investigation to evaluate the effectiveness of the DARS Traffic Plus mobile application to improve driving safety conditions. 

Response:

None.

Action(s) taken:

No action was taken.

Reviewer #1:

Comment R.1.2              The experimental design appears correct and tools and methods are described. However, the two types of analysis are not equally convincing.

Response:

Thank you for your comment.

Action(s) taken:

No action was taken.

Reviewer #1:

Comment R.1.3              The first, based on a questionnaire survey, describes the results completely and comments on them correctly. The second, defined by the authors as quantitative analysis, presents many inaccuracies that require clarification.

Response:

We have made improvements, as presented in reference R1.8, about the importance of the four notification and/or reporting options integrated into the mobile application of DT+. We have rewritten parts of the manuscript for better readability and improved and clarified parts of the manuscript that the authors define as quantitative analysis.

Action(s) taken:

We have made improvements, as presented in reference R1.8, about the importance of the four notification and/or reporting options integrated into the mobile application of DT+. We have rewritten parts of the manuscript for better readability and improved and clarified parts of the manuscript that the authors define as quantitative analysis.

Reviewer #1:

Comment R.1.4              On the other hand, the inadequate significance of objective investigations and measurements of indirect user actions clearly emerges from a sentence by the authors (line 128) combined with a non-existent bibliographical analysis in this regard.

Response:

We have rewritten the "Related works" chapter for better readability and clarity and emphasise the main points of our research. We have also expanded the statement in line 128 and added relevant research literature references. Open-ended questions and interviews are one of the typical methods of driving simulator research. These methods are combined with the data gathered from driving parameters from the driving simulator and the eye-tracking device installed in the driving simulator.

Action(s) taken:

We have rewritten the "Literature review" chapter, expanded the statement in line 128 of the original manuscript, now the statement is in line 92, and added relevant research literature. (please see line 77, with selected show simple markup )

Reviewer #1:

Comment R.1.5              It is therefore suggested, as a first action, to expand the bibliographic analysis by inserting studies relating to the analysis of risk perception or more generally safety conditions through measures of human driving behavior (kinematic-dynamic and/or psychophysiological measures). 

Response:

As our response in comment R.1.4, We have rewritten the " Literature review" chapter for better readability and clarity, emphasising the main points of our research and expanding it with relevant publications.

Action(s) taken:

We have rewritten and expanded the "Literature review" chapter with relevant publications. (please see line 77, with selected show simple markup)

Reviewer #1:

Comment R.1.6              Concerning the effectiveness assessments through quantitative driving parameters, it is suggested to specify the reference numerical threshold values. It is necessary for the authors to clarify the meaning of "too fast acceleration" or "Driving too slow"... It is not even possible to consider it as a comparative evaluation since the quantities are highly non-homogeneous

Response:

The driving simulator scenarios were designed according to traffic regulations. The participants of the driving simulator study were instructed to drive according to roadside traffic signalisation and traffic regulations. They were instructed not to commit traffic violations. During the driving scenarios, we observed whether participants were driving at the speed limit, below the speed limit or above the speed limit. Due to the driving simulator limitation, we were able to detect if they were driving at the speed limit, below or above it. We have set the 5 km/h interval from the current speed limit to determine how they drive. If they were driving more than 5 km/m above the speed limit, they triggered the "Driving too fast" event. If they were driving more than 5 km/m below the speed limit, they triggered the "Driving too slow" event. The driving simulator parameter "too fast acceleration" was triggered if participants pressed the accelerator pedal to the maximum power position for more than 5 seconds.

Action(s) taken:

We have added clarification of the simulator driving parameters to the text; please see lines 393-402, with selected show simple markup.

Reviewer #1:

Comment R.1.7              Regarding the comparison of results between scenarios (tab.5, tab.6, tab.7) it is necessary to specify the evaluation scale adopted.

Response:

Table 5 compares the p-values of significant differences in quantitative driving parameters for Scenario S1 (without DT+ mobile application) and Scenario S2 (with DT+ mobile application) in the driving simulator. We were searching for the significant differences between the quantitative driving parameters of scenarios S1 and S2.

Table 6 compares the p-values of significant differences in quantitative driving parameters for Scenario S1 (without DT+ mobile application) and Scenario S2 (with DT+ mobile application) in greater detail.  We focused on the short intervals where the participants received or sent reports of hazardous locations events. We looked at the data from two sides. We are comparing the p-values of frequency (how many times) specific events occurred (Table 6 a)) and the duration of time spent in intervals that they saw reports or sent reports (Table 6 b)).

Table 7 presents the actual data on how many times the participants saw the notifications, how many buttons they pressed and how many notifications they sent themselves. The results for scenarios with the DT+ mobile application are presented in the Table 7. Scenario S1 did not include the support of a mobile application in the scenario's design and is thus not presented in the table. Scenario S3 was different from scenario S2, as in scenario S3, where we tested additional notification functionalities.

Action(s) taken:

In Table 7, we have marked scenario S2 side of the table as Table 7a) and the scenario S3 side as Table 7b). Additionally, we have entered the maximum value of the parameter in brackets, as reported in lines 464-469 of the original manuscript. The new table is located in line 471, with selected show simple markup.

Reviewer #1:

Comment R.1.8              The meaning of the rating, the results of which are reported in table 12, is absolutely unclear. It is suggested to explain how the ratings can be made comparable.

Response:

Participants were asked to rate the importance of the four notification and/or reporting options integrated into the mobile application of DT+: Reporting Poor Visibility, Obstacle on the road followed by Traffic Jam Ahead and Accident Zone. They were asked which hazardous location notification was the most important for them and would like to get notified about them. The rating was done on a 5-point  Likert scale, where a value of 1 meant "not important" and a value 5 meant "very important". The results show the mean value results for all 33 participants.

Action(s) taken:

We added text about the 5-point Likert scale used for evaluating the importance of Reporting Poor Visibility, Obstacle on the road followed by Traffic Jam Ahead and Accident Zone events on the road. We also added a short description of why rating the events were significant. Please see lines 594-609, with selected show simple markup.

Reviewer #1:

Comment R.1.9              Finally, in relation to the Time distributions of viewing of area of interests, it is suggested to specify any correlations with the motion conditions, or other specific literature indicators to evaluate the levels of understanding and/or attention.

Response:

Thank you for your suggestion.

Action(s) taken:

We have added Figure 4 (line 476) to better represent the Area of interest in the driving simulator. We have added literature sources to the related works chapter "Literature review", line 77, with selected show simple markup.

Reviewer 2 Report

Comments and Suggestions for Authors

The background Literature Review really jumps around on topics, coming across more like point form notes on a variety of topics, rather than drawing focus to specifics this study intends to address. The paragraph on VIMS being a good example, that comes out of the blue. 

The results sections similarly presents a lot of data without really conveying useful information for the reader. What defines "driving too fast", "too slow", "accelerating too fast" etc.? And how would the app be expected to impact them? The one noted metric that changed with app usage was a reduction in driving "too slow". But considering this is in an area with a hazard, is slower not more desirable?

More information also required on specifics of questionnaire and associated instructions. I don't see how items such as Usability, Intention to Use, or Product Loyalty were evaluated in Scenario 1 with no mobile application in use. Usability of what??

Comments on the Quality of English Language

Some minor edits required (e.g. lover levels vs lower), okay overall.

Author Response

Letter to Editor and Reviewers-3090723

Editor's and reviewers' comments analysis

Manuscript ID: sensors-3090723-R2

Type: Article

Title: Enhancing Driving Safety Through User Experience Evaluation of the C-ITS Mobile Application: A Case Study of the DARS Traffic Plus App in a Driving Simulator Environment

Authors: Gregor Burger, Jože Guna

This document contains the list of respected editor's and reviewers' comments on the manuscript " Enhancing Driving Safety Through User Experience Evaluation of the C-ITS Mobile Application: A Case Study of the DARS Traffic Plus App in a Driving Simulator Environment ", submitted to the Sensors Journal. We have found the comments very valuable. This document also contains the detailed authors' responses and explanations of changes done according to the editor's and reviewers' comments. When checking the line numbers in the article please use the "Show simple markup" Word view. All the reviewers' comments and suggestions were addressed.

Reviewers' Comments and Authors' Responses

Reviewer #2:

The background Literature Review really jumps around on topics, coming across more like point form notes on a variety of topics, rather than drawing focus to specifics this study intends to address. The paragraph on VIMS being a good example, that comes out of the blue. 

The results sections similarly presents a lot of data without really conveying useful information for the reader. What defines "driving too fast", "too slow", "accelerating too fast" etc.? And how would the app be expected to impact them? The one noted metric that changed with app usage was a reduction in driving "too slow". But considering this is in an area with a hazard, is slower not more desirable?

More information also required on specifics of questionnaire and associated instructions. I don't see how items such as Usability, Intention to Use, or Product Loyalty were evaluated in Scenario 1 with no mobile application in use. Usability of what??

Reviewer #2:

Comment R.2.1              The background Literature Review really jumps around on topics, coming across more like point form notes on a variety of topics, rather than drawing focus to specifics this study intends to address. The paragraph on VIMS being a good example, that comes out of the blue. 

Response:

We have rewritten the "Literature review" chapter for better readability and clarity and emphasise the main points of our research.

Action(s) taken:

We have rewritten the "Literature review" chapter and added relevant research literature, line 77, with selected show simple markup.

Reviewer #2:

Comment R.2.2              The results sections similarly presents a lot of data without really conveying useful information for the reader. What defines "driving too fast", "too slow", "accelerating too fast" etc.? And how would the app be expected to impact them? The one noted metric that changed with app usage was a reduction in driving "too slow". But considering this is in an area with a hazard, is slower not more desirable?

Response:

The driving simulator scenarios were designed according to traffic regulations. The participants of the driving simulator study were instructed to drive according to roadside traffic signalisation and traffic regulations. They were instructed not to commit traffic violations. During the driving scenarios, we observed whether participants were driving at the speed limit, below the speed limit or above the speed limit. Due to the driving simulator limitation, we were able to detect if they were driving at the speed limit, below or above it. We have set the 5 km/h interval from the current speed limit to determine how they drove. If they were driving more than 5 km/m above the speed limit, they triggered the "Driving too fast" event. If they were driving 5 more than 5 km/m below the speed limit, they triggered the "Driving too slow" event. The driving simulator parameter "too fast acceleration" was triggered if participants pressed the accelerator pedal to the maximum power position for more than 5 seconds.

Furthermore, our findings are supported by the results of the Ipswich Connected Vehicle Pilot study in Australia [20], [21]. We agree that slower driving in the hazardous zone is more desirable. However, our research results indicate that when drivers use a mobile app as a system for advanced notifications on hazardous events on the road, they adapt their driving style and speed before entering the hazardous event zone or at the beginning of this zone. After that, they keep their newly adapted speed constant, and traffic moves more fluidly. Therefore, they drive at higher average speeds than scenarios without mobile application support but still within traffic regulations. In the case of driving without mobile application notifications support, all the speed adjustments are made at the location of the hazardous event on the road. This causes erratic driving with congestion and less fluid traffic. Therefore lower average speeds.

Action(s) taken:

We have added clarification of the simulator driving parameters to the text; please see lines 393-402 with selected show simple markup.

Reviewer #2:

Comment R.2.3              More information also required on specifics of questionnaire and associated instructions. I don't see how items such as Usability, Intention to Use, or Product Loyalty were evaluated in Scenario 1 with no mobile application in use. Usability of what??

Response:

The user experience evaluation of the DT+ mobile application is done through driving. Scenario S1 is similar to scenario S2, with the support of the DT+ mobile application in scenario S2. Participants rated their whole driving experience with a user experience questionnaire. The difference between the scores of the questionnaires in scenario S1 and scenario S2 is the user experience of the DT+ mobile application. In our case, the scores for the user experience questionnaire were higher for scenario S2. Therefore, the DT+ mobile application brings a positive user experience to driving.

Action(s) taken:

We added a clarification to the text regarding user experience evaluation of the mobile application, please see lines 344-350, with selected show simple markup.

Reviewer 3 Report

Comments and Suggestions for Authors This study did a good job in terms of experiment design and data analysis. However, I do not think the manuscript is suitable for publication without significant changes for the following reasons:
  1. The work presented in the manuscript is a user study with the main purpose of illustrating the effectiveness of a mobile application in changing driving behavior. I don't think this fits particularly well within the scope of the journal.
  2. The core of this study should be illustrating how and why DT+ can positively affect driver behavior. The author focused too much on the evaluation process itself but did not illustrate and explain what design concept, visual component, and alert scheme resulted in these changes. Without a detailed explanation of the design, the findings cannot be generalized to transferrable design principles that would benefit more people.
Specific Comments Line 77 - I do not see a logical connection between the paragraphs in the literature review. For example, why is motion sickness related to this work? Line 107 - "Lower levels of automation" - this phrase needs clarification or expansion to fit contextually within the manuscript. Line 313 - This paragraph is too lengthy and hard to read. Why are some of the demographic details here relevant? Especially the percentages? Are they useful when explaining some of the research findings? Line 308 - For the results section, the author provides too many tables and unnecessary details, making this section hard to read and to grasp important information. Perhaps use more diagrams, discuss significantly distinct pairs in more detail, and very briefly mention insignificant pairs. The SD values are not particularly interesting compared to the p-values. Line 499 - The author needs to use better figures to explain the definition of AOIs and explain the selection of AOIs. Line 538 - How did you quantitatively analyze the eye-tracking data? What are the measures (e.g., fixation duration, scan path length, etc.)? Comments on the Quality of English Language

The manuscript is well-written. The conciseness could be improved though. 

Author Response

Letter to Editor and Reviewers-3090723

Editor's and reviewers' comments analysis

Manuscript ID: sensors-3090723-R3

Type: Article

Title: Enhancing Driving Safety Through User Experience Evaluation of the C-ITS Mobile Application: A Case Study of the DARS Traffic Plus App in a Driving Simulator Environment

Authors: Gregor Burger, Jože Guna

This document contains the list of respected editor's and reviewers' comments on the manuscript " Enhancing Driving Safety Through User Experience Evaluation of the C-ITS Mobile Application: A Case Study of the DARS Traffic Plus App in a Driving Simulator Environment ", submitted to the Sensors Journal. We have found the comments very valuable. This document also contains the detailed authors' responses and explanations of changes done according to the editor's and reviewers' comments. When checking the line numbers in the article please use the "Show simple markup" Word view. All the reviewers' comments and suggestions were addressed.

Reviewers' Comments and Authors' Responses

Reviewer #3:

This study did a good job in terms of experiment design and data analysis. However, I do not think the manuscript is suitable for publication without significant changes for the following reasons:

  1. The work presented in the manuscript is a user study with the main purpose of illustrating the effectiveness of a mobile application in changing driving behavior. I don't think this fits particularly well within the scope of the journal.
  2. The core of this study should be illustrating how and why DT+ can positively affect driver behavior. The author focused too much on the evaluation process itself but did not illustrate and explain what design concept, visual component, and alert scheme resulted in these changes. Without a detailed explanation of the design, the findings cannot be generalised to transferrable design principles that would benefit more people.

Specific Comments Line 77 - I do not see a logical connection between the paragraphs in the literature review. For example, why is motion sickness related to this work? Line 107 - "Lower levels of automation" - this phrase needs clarification or expansion to fit contextually within the manuscript. Line 313 - This paragraph is too lengthy and hard to read. Why are some of the demographic details here relevant? Especially the percentages? Are they useful when explaining some of the research findings? Line 308 - For the results section, the author provides too many tables and unnecessary details, making this section hard to read and to grasp important information. Perhaps use more diagrams, discuss significantly distinct pairs in more detail, and very briefly mention insignificant pairs. The SD values are not particularly interesting compared to the p-values. Line 499 - The author needs to use better figures to explain the definition of AOIs and explain the selection of AOIs. Line 538 - How did you quantitatively analyse the eye-tracking data? What are the measures (e.g., fixation duration, scan path length, etc.)?

Comments on the Quality of English Language

The manuscript is well-written. The conciseness could be improved though. 

Reviewer #3:

Comment R.3.1    This study did a good job in terms of experiment design and data analysis. However, I do not think the manuscript is suitable for publication without significant changes for the following reasons:

Response:

Thank you for your comment.

Action(s) taken:

No action was taken.

Reviewer #3:

Comment R.3.2    1.         The work presented in the manuscript is a user study with the main purpose of illustrating the effectiveness of a mobile application in changing driving behavior. I don't think this fits particularly well within the scope of the journal.

Response:

We believe the manuscript is relevant to the scope of the HCI topic of the journal. The DT+ application does influence driving behaviour, and that is true. However, our research was broader, and we wanted to examine how participants were interacting with the mobile application. Did they understand what information the notifications on the screen of mobile applications were reporting to them? We researched not only receiving hazardous notifications but also how participants interact with mobile application when reporting the hazardous notifications locations on the road. In addition to visual interfaces, we examined sound notifications. Our research also includes a lot of quantitave data based results, such as driving simulator data and eye-tracking data, thus falling within the scope of Sensors journal.

Action(s) taken:

We have rewritten the "Literature review" chapter to show the relevance of the manuscript in the HCI field, line 77, with selected show simple markup.

Reviewer #3:

Comment R.3.3    2.             The core of this study should be illustrating how and why DT+ can positively affect driver behavior. The author focused too much on the evaluation process itself but did not illustrate and explain what design concept, visual component, and alert scheme resulted in these changes. Without a detailed explanation of the design, the findings cannot be generalised to transferrable design principles that would benefit more people.

Response:

We added a screenshot of the Hazardous Location Notification, more precisely for an accident on the road. The notification is shown from the top of the mobile phone screen. The colour of the background of the notification is red, meaning the accident location is nearby. Notification has an icon indicating the car accident, location information, distance to the location of the accident and the expected time delay for the driver.

The design of the user interface is important for presenting the information of the notifications. However, we want to emphasize that the main point of our research was the evaluation of the user experience of driving with and without the DT+ mobile application.

Action(s) taken:

We added a description of how the hazardous location notifications were shown in the application and reported by the study participants; please see lines 557-567, with selected show simple markup. We inserted a new Figure 9 of the DT+ mobile application hazardous location notifications, see line 610, with selected show simple markup.

Reviewer #3:

Comment R.3.4    Line 77 - I do not see a logical connection between the paragraphs in the literature review. For example, why is motion sickness related to this work?

Response:

We have rewritten the "Literature review"  hapter for better readability and clarity and emphasise the main points of our research.

Action(s) taken:

We have rewritten the "Literature review" chapter and added relevant research literature, line 77, with selected show simple markup.

Reviewer #3:

Comment R.3.5    Line 107 - "Lower levels of automation" - this phrase needs clarification or expansion to fit contextually within the manuscript.

Response:

We have rewritten the "Literature review" chapter for better readability and clarity and emphasise the main points of our research. The phrase has been removed to improve readability.  

Action(s) taken:

We have rewritten the "Literature review" chapter and added relevant research literature, line 77, with selected show simple markup.

Reviewer #3:

Comment R.3.6    Line 313 - This paragraph is too lengthy and hard to read. Why are some of the demographic details here relevant? Especially the percentages? Are they useful when explaining some of the research findings?

Response:

We have rewritten the paragraphs 4.1, Participant demographics data and 4.2. Technology and mobile applications usage.

Action(s) taken:

We have rewritten the paragraphs 4.1, Participant demographics data and 4.2. Technology and mobile applications usage with less focus on percentages, line 308, with selected show simple markup.

Reviewer #3:

Comment R.3.7    Line 308 - For the results section, the author provides too many tables and unnecessary details, making this section hard to read and to grasp important information. Perhaps use more diagrams, discuss significantly distinct pairs in more detail, and very briefly mention insignificant pairs. The SD values are not particularly interesting compared to the p-values.

Response:

We understand that data is presented in extensive tables as we wish to substantiate our findings on measurable quantitative data, such as the driving parameters from the driving simulator. We believe the SD values are essential in this case as the aggregated data is presented for all 33 participants for scenarios S1 and scenario S2. Furthermore, we were looking for the significant differences between the driving parameters, first on the aggregated level and second on the more detailed level of notification and reporting intervals. In the text, we have furthermore highlighted the driving parameters that had significant differences between the driving parameters for scenario S1 and scenario S2, for better readability.  

Action(s) taken:

We made some improvements to the readability of the Results chapter. We rewrote some parts of the text and highlighted the important data in the tables, especially the p-value tables, line 303, with selected show simple markup.

Reviewer #3:

Comment R.3.8    Line 499 - The author needs to use better figures to explain the definition of AOIs and explain the selection of AOIs.

Response:

We have added Figure 4 to represent the Area of interest in the driving simulator. We have added literature sources to the "Literature review", line 77, with selected show simple markup.

Action(s) taken:

We have added Figure 4 (line 476) to better represent the Area of interest in the driving simulator. We have added literature sources to the related works chapter "Literature review", line 92.

Reviewer #3:

Comment R.3.9    Line 538 - How did you quantitatively analyse the eye-tracking data? What are the measures (e.g., fixation duration, scan path length, etc.)?

Response:

Thank you for your suggestion on how to improve the eye-tracking results.

Action(s) taken:

We have expanded the description of eye-tracking measures. Please see line 520-531

Reviewer #3:

Comment R.3.10 The manuscript is well-written. The conciseness could be improved though.

Response:

Thank you for your kind comment about the preparation of the manuscript.

Action(s) taken:

No action was taken.

Round 2

Reviewer 1 Report

Comments and Suggestions for Authors

The authors interpreted and consequently analyzed the Quantitative driving parameters incorrectly, attributing a value to the frequency of occurrence of these parameters. Evaluating the frequency of certain maneuvers is certainly a quantitative evaluation, however the definition of these parameters is strictly qualitative, although, as stated by the authors... “according to roadside traffic signaling and traffic regulations”.

The simple reference to the speed threshold exceeded, 5 km/h (km/m ???), cannot be accepted for quantitative analysis. However, as confirmed by the literature [*] the evaluation appears to be quantitative if made with respect to individual speed values, even above or below threshold, or to distributions thereof.

….. We have set the 5 km/h interval from the current speed limit to determine how they drive. If they were driving more than 5 km/m above the speed limit, they triggered the "Driving too fast" event. If they were driving more than 5 km/m below the speed limit, they triggered the "Driving too slow" event.

Similarly, as regards acceleration, the time for which one decelerates or accelerates is of little significance if this is not correlated to the intensity of pressure on the relevant pedal and therefore to the acceleration or deceleration value. This observation is also widely supported by literature studies.

De Blasiis M.R., Ferrante C., Veraldi V. (2020) Driving Risk Assessment Under the Effect of Alcohol Through an Eye Tracking System in Virtual Reality. In: Arezes P. (eds) Advances in Safety Management and Human Factors. AHFE 2019. Advances in Intelligent Systems and Computing, vol 969. pp. 329–341 Springer Nature Switzerland AG 2020 https://doi.org/10.1007/978-3-030-20497-6_31

Li, S., Blythe, P., Zhang, Y. et al. Analysing the effect of gender on the human–machine interaction in level 3 automated vehicles. Sci Rep 12, 11645 (2022). https://doi.org/10.1038/s41598-022-16045-1

M.R. De Blasiis, S. Diana, V. Veraldi (2018) Safety audit for weaving maneuver: A driver simulation safety analysis, Journal of Transportation Safety & Security, 10:1-2, 159-175, DOI: 10.1080/19439962.2017.1323060  

In conclusion, if it is not possible to integrate the paper with the suggested analyses, its validity is confirmed for publication as a questionnaire survey related to investigations into driver behavior

Reviewer 3 Report

Comments and Suggestions for Authors

The author descently addressed the reviewer's comments.